# The Surgical Timing and Complications of Rib Fixation for Rib Fractures in Geriatric Patients

**DOI:** 10.3390/jpm12101567

**Published:** 2022-09-23

**Authors:** Szu-An Chen, Chien-An Liao, Ling-Wei Kuo, Chih-Po Hsu, Chun-Hsiang Ouyang, Chi-Tung Cheng

**Affiliations:** Department of Trauma and Emergency Surgery, Chang Gung Memorial Hospital, Linkou, Chang Gung University, Taoyuan 333, Taiwan

**Keywords:** rib fracture, rib fixation, geriatric, chest trauma

## Abstract

Rib fractures (RF) are a common injury that cause significant morbidity and mortality, especially in geriatric patients. RF fixation could shorten hospital stay and improve survival. The aim of this retrospective study was to evaluate the clinical impact and proper surgical timing of RF fixation in geriatric patients. We reviewed all the medical data of patients older than 16 years old with RF from the trauma registry database between January 2017 and December 2019 in Chang Gung Memorial Hospital. A total of 1078 patients with RF were enrolled, and 87 patients received RF fixation. The geriatric patients had a higher chest abbreviated injury scale than the non-geriatric group (*p* = 0.037). Univariate analysis showed that the RF fixation complication rates were significantly related to the injury severity scores (Odds ratio 1.10, 95% CI 1.03–1.20, *p* = 0.009) but not associated with age (OR 0.99, 95% CI 0.25–3.33, *p* = 0.988) or the surgical timing (OR 2.94, 95% CI 0.77–12.68, *p* = 0.122). Multivariate analysis proved that only bilateral RF was an independent risk factor of complications (OR 6.60, 95% CI 1.38–35.54, *p* = 0.02). RF fixation can be postponed for geriatric patients after they are stabilized and other lethal traumatic injuries are managed as a priority.

## 1. Introduction

Rib fractures (RF) after chest wall trauma are a common injury [1]. However, RF could lead to significant morbidity, mortality, and even long-term disabilities [2,3]. Previous treatment options were limited to conservative management such as analgesia, pulmonary toilet, and oxygen support in order to wait enough time for wound healing [4]. Meanwhile, some investigations found that stabilization of the chest wall would contribute to decreasing the length of hospital and intensive care unit (ICU) stay among RF patients [5,6,7]. Currently, there is growing evidence supporting the idea that RF fixation helps to improve patients’ survival and accelerate the time of return to a normal functional state. RF fixation was also thought to have promising benefits in terms of less narcotics use, the avoidance of tracheostomy, and a better quality of life [7,8].

Geriatric patients who are over 65 years old have been well documented to have a poor prognosis of RF with higher mortality and pneumonia rates than other age groups [2,9,10]. Nevertheless, there have been few studies dedicated to investigating the impact of RF fixation on the geriatric population [4]. Although some investigations demonstrated improvement in mortality and pneumonia rates after RF fixation [7,8], these data were from relatively small case numbers and did not analyze the outcomes of the senior population specifically. With the development of plating systems for RF fixation and chest wall stabilization, the practice paradigm for RF management is shifting, and operative intervention is available at present [11]. Unlike young adults, geriatric patients with chronic disorders elevated the surgical risk to decrease the benefit of RF fixation [2,11,12]. Furthermore, the absolute surgical timing [13,14,15] and selection criteria [6,15,16,17] for this procedure remained uncertain. Previous literature reviews recommended early surgical fixation within 3 days after injury [18]. However, traumatic patients usually had more fatal diseases that needed to be dealt with as a priority, which postponed RF fixation after 3 days. In this study, we aimed to evaluate the clinical impact and proper surgical timing of RF fixation in geriatric patients.

## 2. Materials and Methods

### 2.1. Study Design and Case Definition

We collected prospective data using a trauma registry at Chang Gung Memorial Hospital (CGMH), Linkou, Taiwan, which is a Level I trauma center in Taiwan that treats approximately 3500 trauma patients annually. Demographic data, medical history, perioperative management, treatment outcomes, and information of complications were prospectively recorded into a computerized database. We retrospectively reviewed all patients older than 16 years old with RF from the trauma registry database between January 2017 and December 2019 in CGMH. Case enrollment was according to the International Classification of Diseases, tenth Revision, Clinical Modification (ICD-10-CM) codes (901.0, 901.2, 902.0, and 902.1). This study was approved by the Internal Review Board of CGMH (202001578B0). Our team also used the same database and published another topic entitled “The Feasibility and Efficiency of Remote Spirometry System on the Pulmonary Function for Multiple Ribs Fracture Patients” [19].

All participants enrolled in this study were managed by a trauma team when they arrived at the emergency room through discharge from the hospital. They also received outpatient department follow-up after discharge. Patients without a definitive diagnosis of rib fracture or with a lost follow-up were excluded from our study. Participants above 65 years old belonged to the geriatric group, and those younger 65 years old were the non-geriatric group. Body mass index (BMI) > 27 was classified as obesity according to the definition from the Health Promotion Administration in Taiwan. We collected the demographic data and severity evaluation scores including the revised trauma score (RTS), abbreviated injury scale (AIS), injury severity score (ISS), new injury severity score (NISS), and trauma injury severity score (TRISS). We used chest computed tomography for the detection of RF. Both the preoperative and postoperative chest plain films were collected for further analysis (Figure 1).

Preoperative evaluations such as fractured rib number, location, associated pulmonary complications, therapeutic procedures, and surgical timing after trauma were all reviewed. Postoperative conditions including recovery, complications, length of ICU stay (ICU LOS), length of hospital stay (HLOS), necessary tracheostomy, sequelae, and mortality were recorded.

Multiple ribs fracture is defined as at least three displaced rib fractures. We used oral analgesics with acetaminophen plus long-acting non-steroidal anti-inflammatory drugs for the initial pain control therapy. Morphine injection would be given every 4 h if needed. If a patient asked for morphine injection for more than three doses in one day, oral opioid medication would be added. We further considered intercostal nerve block for pain control if the visual analogue scale of the pain score was persistently at least four or if morphine infection was still required for more than three doses in one day. Patient-controlled analgesia was the last step when all the pain management strategies failed. Flail chest and multiple displaced RF with a failure of pain control were the two indications of RF fixation in this study. We performed an exploratory thoracoscope to evaluate the pleural cavity and then fixed and plated the ribs. The early RF fixation group was defined as receiving operative intervention within 5 days, and the deferred group patients underwent RF fixation 5 days after RF. The RF fixation was performed by the same team using the same protocols.

### 2.2. Statistical Analysis

Pearson’s *X*_2_ test and Fisher’s exact test were used to compare categorical variables. Quantitative variables were compared by the Mann–Whitney U test. The risk factor analysis of complication was performed using univariate logistic regression and backward stepwise logistic regression. All statistical analyses were performed with R 3.6.3 with the packages “tableone”, “finalfit”, and “MASS”. A value of *p* < 0.05 was considered statistically significant.

## 3. Results

### 3.1. Demographic Data

During the study period, a total of 1078 patients with RF were admitted to our hospital. Overall, 87 patients received RF fixation, and the other patients underwent conservative management (Table 1). The study diagram is presented in Figure 2. 

Males accounted for 71.3% (*n* = 62) of the participants, and 32.2% (*n* = 28) were obese. The overall median age was 56 years. The causes of injury included motor vehicle collisions (66.7%, *n* = 58), falls (31.0%, *n* = 27), and crushing (2.3%, *n* = 2). The median scores of ISS, NISS, and TRISS were 20, 24, and 0.94, respectively. A total of 12.6% (*n* = 11) of patients had bilateral rib fractures. The median number of fractured ribs was 6, and 49 patients had flail segment fracture at arrival. There were 25 geriatric patients (28.7%), with a median age of 70 years old. Compared with the non-geriatric group, the geriatric patients presented with higher chest AIS (*p* = 0.037). The geriatric patients also had higher rates of rib fracture with flail segments than the non-geriatric group (76% versus 48.4%, *p* = 0.031), which might be related to the osteoporosis of aging. RF fixation was also performed in some severe-head-injury patients, and 16.1% (*n* = 14) of patients had a head AIS of 3.

### 3.2. The Clinical Effect of Rib Fracture Fixation

The median number of days to receive RF fixation was 5.1 days. The majority of participants had early RF surgery. A total of 16 (64%) geriatric patients underwent early surgery within 5 days, while others received late surgery 5 days after RF. There was no mortality among geriatric patients. Only one non-geriatric patient (1.15%) died due to sepsis with multiorgan failure one month after RF fixation. The median hospital length of stay was 12 days, with no significant difference between the geriatric and non-geriatric groups (*p* = 0.454). Overall, 39.1% of patients need ICU admission. Geriatric patients had higher ICU admission rates, but the ICU length of stay and the complication rates were similar to those of the non-geriatric group. (Table 2)

### 3.3. Risk Factors Associated with Postoperative Complications

A total of 14 (16.1%) patients had postoperative complications, including 9 (64%) with hospital-acquired pneumonia, 1 (7%) with pneumothorax, 1 (7%) with intractable pain, 1 (7%) with wound infection, and 2 (15%) with cerebral vascular accidents. The risk factors analysis of complications is shown in Table 3. Univariate analysis showed that the ISS, the NISS, the TRISS, the number of fractured ribs, the number of fracture sites, and bilateral RF were significant risk factors. On the other hand, age (OR 0.99, 95% CI 0.25–3.33, *p* = 0.988) and surgical timing (OR 2.94, 95% CI 0.77–12.68, *p* = 0.122) did not increase the risk of postoperative complications. Further backward stepwise regression showed that only bilateral RF was an independent risk factor of complications (OR 6.60, 95% CI 1.38–35.54, *p* = 0.02).

## 4. Discussion

Early RF fixation could reduce complications, improve pulmonary function, and shorten the HLOS. However, severe trauma-associated injuries usually limit the possibility of early operative intervention for RF. In this study, we provided evidence that delayed RF fixation after 5 days did not lead to a poorer outcome in geriatric patients. No patient needed further tracheostomy or other ventilation assistance after RF fixation. Some publications consistently found that age was not a risk factor for poor prognosis after RF fixation [20]. 

In the past decades, a minority of surgeons recognized that selected patients with flail chests received a benefit from surgical fixation. However, only sporadic series of RF fixation utilizing various techniques were reported [21]. Patients with chest wall deformities were also considered as another indication for fixation if the displaced RF or chest wall defect was too severe to heal spontaneously [22]. With the advances of technology and material science, increasing surgical instruments and plates were invented to change the ways to deal with this trauma. RF fixation has become an available therapeutic option with multiple clinical advantages for severe RF patients [5,23,24,25]. Surgical fixation of the fractured ribs also ensured the restoration of the chest wall integrity and reduced chest pain, contributing to the recovery in pulmonary function, which was necessary for adequate ventilation and the effective clearance of airway secretions [6,26].

Nevertheless, associated traumatic injuries and underlying chronic disorders increased the surgical risk in the geriatric population [2,12]. Some previous research even excluded the elders in the study design to prevent further complications of RF fixation due to potential osteoporosis [8]. Currently, more and more publications provide evidence to support early RF fixation for geriatric patients [4,27]. Surgical fixation contributed to the earlier recovery of pulmonary function [4]. RF fixation has also been shown to decrease the long-term sequelae of RF such as chronic pain, chest wall stiffness, impaired breathing excursions, and chest wall deformity [28,29,30]. In addition, thoracoscopic surgery combined with RF fixation had several advantages for severe chest trauma patients [31]. Therefore, our team also applied a thoracoscope to evacuate the hemothorax and pleural effusion during RF fixation to reduce the possibility of lung atelectasis and defer residual pleural effusion.

The best surgical timing of RF fixation has been discussed for years, and early intervention was recommended [15,27]. However, other associated traumatic injuries usually postponed the timing for RF fixation. Delayed operation was thought to prolong ICU admission, increase ventilator-dependent days, and lead to complications such as pneumonia, lung atelectasis, residual pleural effusion, and even mortality [32]. In this study, we found that the main reasons for prolonged ventilator-dependent days and ICU LOS were related to other traumatic-associated injuries rather than RF fixation complications. Therefore, the most appropriate time for RF fixation should be when the patients received primary resuscitation and achieved stabilization.

There were some limitations in our study. First, the mechanism of the bone healing process was complicated. Some studies found that the irisin hormone levels and the high mean platelet volume were associated with bone fracture [33,34]. Fractures in patients with high mean platelet volume values were more likely to need additional surgical interventions [34]. However, we did not evaluate all the associated predictors but focused mostly on the influence of age and the severity of trauma. Further investigations are still needed in the future. Second, this retrospective observational study was performed in one single medical center. The surgical timing was based on the patient’s clinical condition rather than randomly assigned, which might cause selection bias. Third, the RF fixation devices were not covered by the National Health Insurance during the study period. Patients would refuse this surgery because it was unaffordable. Therefore, the number of participants was relatively small for detecting significant differences in all the variables used in the study.

## 5. Conclusions

RF fixation is an appropriate therapeutic option in both geriatric and young RF patients to improve pulmonary function, relieve pain, and maintain life quality. The surgical timing was not a significant risk factor for increasing complication rates or worsening clinical outcome after RF fixation in geriatric patients. Therefore, RF fixation can be postponed for geriatric patients after they are stabilized and other lethal traumatic injuries are managed as a priority.

## Figures and Tables

**Figure 1 jpm-12-01567-f001:**
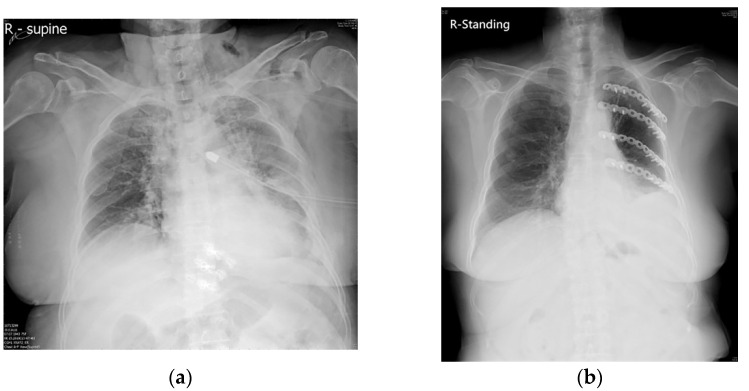
The posteroanterior chest plain films of the multiple ribs fracture of one geriatric patient. (**a**) Geriatric patient suffering from left multiple ribs fracture with hemothorax and chest tube insertion. (**b**) After rib fixation with plates, the thoracic cage was reconstructed and the lung expanded well.

**Figure 2 jpm-12-01567-f002:**
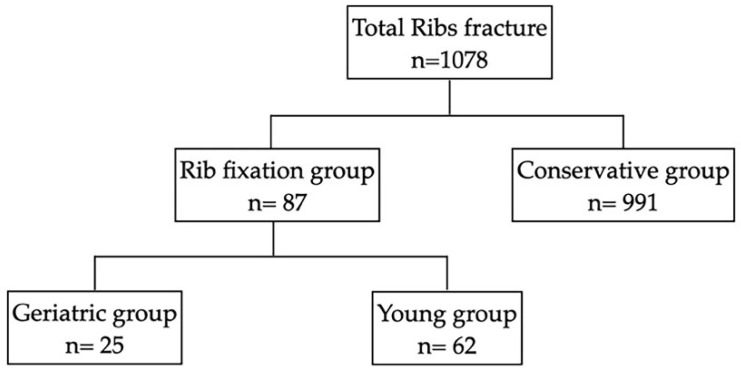
The study diagram of the enrollment of rib fracture patients.

**Table 1 jpm-12-01567-t001:** The overall demographics and a comparison between the geriatric population and the non-geriatric population (*n* = 87).

	Total	Non-Geriatric	Geriatric	*p*-Value †
	(*n* = 87)	(*n* = 62, 71.3%)	(*n* = 25, 28.7%)	
**Demographic data**				
Gender				0.794
Male (*n*, %)	62, 71.3%	45, 72.6%	17, 68.0%	
Female (*n*, %)	25, 28.7%	17, 27.4%	8, 32.0%	
Age (median, IQR)	56.00 [51.00, 66.00]	54.00 [47.00, 57.75]	70.00 [67.00, 73.00]	<0.001 *
Body weight (median, IQR)	68.00 [60.00, 74.85]	70.00 [60.12, 77.93]	65.00 [58.00, 70.50]	0.117
BMI (median, IQR)	25.30 [22.75, 27.65]	25.40 [22.40, 28.32]	25.10 [23.20, 26.20]	0.318
Obesity (*n*, %)	28, 32.2%	23, 37.1%	5, 20.0%	0.138
Mechanism				0.898
Traffic accident (*n*, %)	58, 66.7%	40, 64.5%	18, 72.0%	
Fall (*n*, %)	27, 31.0%	20, 32.3%	7, 28.0%	
Blunt injury(*n*, %)	2, 2.3%	2, 3.2%	0, 0.0%	
ISS (median, IQR)	20.00 [16.00, 29.00]	20.00 [16.00, 28.50]	20.00 [16.00, 29.00]	0.292
NISS (median, IQR)	24.00 [16.50, 34.00]	24.00 [17.00, 33.50]	24.00 [16.00, 34.00]	0.880
TRISS (median, IQR)	0.94 [0.89, 0.97]	0.96 [0.90, 0.99]	0.92 [0.85, 0.94]	0.010 *
Head AIS ≥ 3 (*n*, %)	14, 16.1%	9, 14.5%	5, 20.0%	0.532
Chest AIS (median, IQR)	4.00 [4.00, 4.00]	4.00 [3.00, 4.00]	4.00 [4.00, 4.00]	0.037 *
Non-chest highest AIS (median, IQR)	2.00 [1.00, 3.00]	2.00 [1.00, 3.00]	2.00 [1.00, 3.00]	0.798
**Trauma pattern**				
Rib fracture with flail segments				0.031 *
Yes (*n*, %)	49, 56.3%	30, 48.4%	19, 76.0%	
No (*n*, %)	38, 43.7%	32, 51.6%	6, 24.0%	
Lung contusion (*n*, %)	40, 46.0%	27, 43.5%	15, 60.0%	0.488
Pneumothorax (*n*, %)	36, 41.4%	26, 41.9%	10, 40.0%	1.000
Hemothorax (*n*, %)	79, 90.8%	54, 87.1%	25, 100.0%	0.099
Bilateral rib fractures (*n*, %)	11, 12.6%	8, 12.9%	3, 12.0%	1.000
No. of fracture ribs (median, IQR)	6.00 [5.00, 8.00]	6.00 [5.00, 8.00]	7.00 [5.00, 8.00]	0.278
No. of fracture sites (median, IQR)	8.00 [5.00, 11.00]	7.00 [5.00, 10.00]	10.00 [6.00, 13.00]	0.049 *
Pre-op ventilator usage (*n*, %)	14, 16.1%	10, 16.1%	4, 16.0%	1.000
Post-op ventilator usage (*n*, %)	33, 37.9%	22, 35.5%	11, 44.0%	0.474
Injury to operation (days, median, IQR)	5.00 [3.00, 6.00]	5.00 [3.00,6.00]	5.00 [3.00,6.00]	0.581

† statistical comparison between non-geriatric ground and geriatric group. * statistical significance. BMI, body mass index; ISS, injury severity score; NISS, new injury severity score; TRISS, trauma injury severity score; AIS, abbreviated injury scale; IQR, interquartile range; ICU, intensive care unit.

**Table 2 jpm-12-01567-t002:** The overall outcome of rib fracture fixation surgery in the geriatric and non-geriatric groups (*n* = 87).

	Total	Non-Geriatric	Geriatric	*p* Value †
	(*n* = 87)	(*n* = 62, 71.3%)	(*n* = 25, 28.7%)	
Surgical timing				1
Early surgery (*n*, %)	56, 64.4%	40, 64.5%	16, 64.0%	
Late surgery (*n*, %)	31, 35.6%	22, 35.5%	9, 36.0%	
Mortality (*n*, %)	1, 1.1%	1, 1.6%	0, 0.0%	1
Length of stay (days, median, IQR)	12.00 [10.00, 15.00]	11.00 [9.25, 14.00]	12.00 [10.00, 17.00]	0.454
ICU admission				0.630
Yes (*n*, %)	34, 39.1%	23, 37.1%	11, 44.0%	
No (*n*, %)	53, 60.9%	39, 62.9%	14, 56.0%	
ICU length of stay (days, median, IQR)	5.00 [3.00,10.00]	7.00 [3.00, 10.00]	3.00 [2.00, 10.00]	0.745
Ventilatory days (days, median, IQR)	4.00 [2.00,7.00]	4.00 [1.00, 13.00]	3.00 [1.00, 8.00]	0.628
Chest tube days (days, median, IQR)	7.00 [4.00,11.75]	7.00 [4.00, 11.75]	9.00 [4.00, 12.00]	0.895
Complications (*n*, %)	14, 16.1%	10, 16.1%	4, 16.0%	1

† statistical comparison between the non-geriatric group and geriatric group. ICU, intensive care unit; IQR, interquartile range.

**Table 3 jpm-12-01567-t003:** The risk factors regarding complications with univariate and multivariate regression analyses.

	No Complication (*n* = 73)	With Complication (*n* = 14)	Univariate OR (95% CI, *p*-Value)	Multivariate † OR (95% CI, *p*-Value)
Age group				
Geriatric (*n*, %)	21, 28.8%	4, 28.6%	0.99 (0.25–3.33, *p* = 0.988)	
Non-geriatric (*n*, %)	52, 71.2%	10, 71.4%	1	
Gender				
Male (*n*, %)	20, 27.4%	5, 35.7%	1.47 (0.41–4.82, *p* = 0.530)	
Female (*n*, %)	53, 72.6%	9, 64.3%	1	
Obesity	25, 89.3%	3, 10.7%	0.52 (0.11–1.86, *p* = 0.353)	
ISS (median, IQR)	20.00 [16.00, 24.00]	29.00 [21.25, 34.00]	1.12 (1.04–1.21, *p* = 0.003 *)	
NISS (median, IQR)	24.00 [16.00, 29.00]	33.50 [24.25, 41.00]	1.09 (1.03–1.16, *p* = 0.003 *)	1.06 (1.00–1.13, *p* = 0.071)
TRISS (median, IQR)	0.96 [0.92, 0.98]	0.85 [0.65, 0.91]	0.01 (0.00–0.23, *p* = 0.010 *)	
No. of fracture ribs (median, IQR)	6.00 [5.00, 8.00]	8.00 [6.25, 11.75]	1.26 (1.05–1.52, *p* = 0.014 *)	
No. of fracture sites (median, IQR)	7.00 [5.00, 11.00]	10.50 [7.50, 14.75]	1.14 (1.02–1.29, *p* = 0.024 *)	
Bilateral rib fractures	5, 45.5%	6, 54.5%	10.20 (2.55–43.63, *p* = 0.001 *)	6.60 (1.38–35.54, *p* = 0.020 *)
Head AIS ≥ 3	10, 71.4%	4, 28.6%	2.52 (0.60–9.29, *p* = 0.176)	
Late surgery	23, 74.2%	8, 25.8%	2.90 (0.91–9.74, *p* = 0.074)	2.94 (0.77–12.68, *p* = 0.122)

* statistical significance. † statistical comparison between no complication ground and complication group.

## Data Availability

The data are available from the corresponding author upon reasonable request.

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
