# Peer review of "The Surgical Timing and Complications of Rib Fixation for Rib Fractures in Geriatric Patients"

_jpm, 2022, doi:10.3390/jpm12101567_

Round 1
Reviewer 1 Report
This retrospective observational study seeks to document the timing and clinical impact management of rib fractures by rib fixation in geriatric patients. The key message is that rib fixation complications were associated with ISS and number of ribs fractured and fracture site, but not age or surgical timing, and that rib fixation can be postponed in geriatric patients.
Strengths: The study shows outcomes that come from rib fixation in the elderly, and delayed rib fixation does not have significant adverse effects. This is a category that is usually overlooked.
Weaknesses: Single center retrospective observational study. Does not indicate if there were any significant co-morbidities in the cohort examined. Multiple biases possible due to selection pressures from clinicians and patients on basis of subjectivity and affordability.
Overall: A fair paper that sheds even more light that rib fixation can improve outcomes even in elderly patients performed after 5 days.
Comments:
1. P3L88 How do the authors define “multiple” rib fractures? What is the number that meets this criteria?
2. Can the authors explain what was the usual analgesia protocol used in the study for all patients? Blocks, patient controlled analgesia?
Author Response
- P3L88 How do the authors define “multiple” rib fractures? What is the number that meets this criteria?
Response: Thank you for this important question. We defined multiple rib fractures as at least 3 displaced ribs fracture. We also added the definition in the methods section.
- Can the authors explain what was the usual analgesia protocol used in the study for all patients? Blocks, patient controlled analgesia?
Response: Thank you very much for your important recommendation. We used oral analgesics with acetaminophen plus long-acting non-steroidal anti-inflammatory drug for initial pain control therapy. Morphine injection would be given every 4 hours if needed. If a patient asked for morphine injection for more than three doses in one day, oral opioid medication would be added. We further considered intercostal nerve block for pain control if the visual analogue scale of pain score was persistent to be at least four or morphine infection was still required for more than three doses in one day. Patient controlled analgesia was the last step when all the pain management strategies failed. We added the description of our pain control policy in the methods section.
Reviewer 2 Report
This is a relatively good study. However, a major revision is still needed. Some suggestions are given as belows:
1. Results section should be more concise. The parameters of predicting surgical effect should be further analyzed, including the correlation
2. The results, analysis and discussion should be combined with clinical effect of surgery.
3. What makes your study special? What do you conclude on your study results?
4. This study has many limitations. Mention the strengths and weaknesses of your study in detail.
5. Discussion is improper.
6. Add References below
PMID: 28869754
PMID: 27471388
Author Response
- Results section should be more concise. The parameters of predicting surgical effect should be further analyzed, including the correlation
Response: Thank you very much for your important recommendation. We revised the results section to be more concise and adjusted the subtitle as “ The clinical effect of ribs fracture fixation”
- The results, analysis and discussion should be combined with clinical effect of surgery
Response: Thank you very much for your recommendation. We revised our study’s results, analysis and discussion sections to address more consistent and important information related to the clinical effect of the ribs fracture fixation.
- What makes your study special? What do you conclude on your study results?
Response: The American Association for the Surgery of Trauma suggested that surgical stabilization of rib fractures might be performed within 72hr after injury. Early RF fixation reduced complications, improve pulmonary function, and shortened the HLOS. However, severe trauma-associated injuries usually limit the possibility of early operative intervention for RF. We provided evidence that delayed RF fixation after 5 days did not lead to poorer outcome in geriatric patients. Besides, no patient needed further tracheostomy or other ventilation assistance after RF fixation in our study. Therefore, we conclude that the surgical timing was not a significant risk factor to increase complication rates or worsen clinical outcome after RF fixation in geriatric patients. RF fixation can be postponed for geriatric patients after they are stabilized and other lethal traumatic injuries were managed as a priority.
- This study has many limitations. Mention the strengths and weaknesses of your study in detail.
Response: Surgical benefits in the geriatric population tended to be overlooked in the past. Our study’s strengths are that we provided data showing the surgical outcomes of rib fixation in the elderly, and delayed rib fixation does not have significant adverse effects. The limitations of our study are described in the following. First, the mechanism of the bone healing process was complicated. Some studies found that the level of irisin hormone and high mean platelet volume were associated with a bone fracture. Fractures in patients with high mean platelet volume values were more likely to need additional surgical interventions. However, we didn’t evaluate all of the associated predictors but focused on the influence of age and the severity of trauma. Further investigations are still needed in the future. Second, this retrospective observational study was performed in one single medical center. The surgical timing was based on the patient’s clinical condition rather than randomly assigned which might cause selection bias. Third, RF fixation was a self-paid operative intervention during the study period. Patients would refuse this surgery because they were unaffordable. Therefore, the number of participants was relatively small to detect significant differences in all the variables used in the study. We have also revised the limitations in the discussion section.
- Discussion is improper.
Response: Thanks for your opinion. We revised the discussion part and added the references you suggested.
- Add References below
Response: Thanks for your useful information. We added the references in the discussion section as the following sentences “the mechanism of bone healing process was complicated. Some studies found that the level of irisin hormone and the high mean platelet volume were associated with bone fracture. Fractures in patients with high mean platelet volume values were more likely to need additional surgical interventions.”
Round 2
Reviewer 2 Report
Dear Authors, many thanks for the effort you have been putting in this article; It's concise, well prepared, well written with good conclusions. The limitations of the study are covered in the discussion section, so I have no problem to publish it.